# Coffee Polyphenol, Chlorogenic Acid, Suppresses Brain Aging and Its Effects Are Enhanced by Milk Fat Globule Membrane Components

**DOI:** 10.3390/ijms23105832

**Published:** 2022-05-23

**Authors:** Keiko Unno, Kyoko Taguchi, Tadashi Hase, Shinichi Meguro, Yoriyuki Nakamura

**Affiliations:** 1Tea Science Center, University of Shizuoka, 52-1 Yada, Suruga-ku, Shizuoka 422-8526, Japan; gp1719@u-shizuoka-ken.ac.jp (K.T.); yori.naka222@u-shizuoka-ken.ac.jp (Y.N.); 2Research and Development, Kao Corporation, 2-1-3 Bunka, Sumida-ku, Tokyo 131-8501, Japan; hase.tadashi@kao.com; 3Biological Science Research, Kao Corporation, Akabane, Ichikai-machi, Haga-gun 321-3497, Japan; meguro.shinichi@kao.com

**Keywords:** cerebral cortex, cognitive function, cAMP-responsive element binding (CREB), hippocampus, inflammation, lifespan, repressor element 1-silencing transcription factor (REST), transforming growth factor β (TGF-β)

## Abstract

Mice feed with coffee polyphenols (CPP, chlorogenic acid) and milk fat globule membrane (MFGM) has increased survival rates and helps retain long-term memory. In the cerebral cortex of aged mice, CPP intake decreased the expression of the proinflammatory cytokine TNF-α, and lysosomal enzyme cathepsin B. The suppression of inflammation in the brain during aging was thought to result in the suppression of the repressor element 1-silencing transcription factor (REST) and prevention of brain aging. In contrast, CPP increased the expression of REST, cAMP-responsive element binding (CREB) and transforming growth factor β1 (TGF-β1) in the young hippocampus. The increased expression of these factors may contribute to the induction of neuronal differentiation and the suppression of memory decline with aging. Taken together, these results suggest that CPP increases CREB in the young hippocampus and suppresses inflammation in the old brain, resulting in a preventive effect on brain aging. The endotoxin levels were not elevated in the serum of aged mice. Although the mechanism of action of MFGM has not yet been elucidated, the increase in survival rate with both CPP and MFGM intake suggests that adding milk to coffee may improve not only the taste, but also the function.

## 1. Introduction

With the advent of an aging society, the prevention of aging, especially brain aging, is an important issue not only for improving the quality of life of individuals, but also for social and economic reasons. Using mouse models of aging, we found that the long-term intake of catechins, a green tea polyphenol, inhibits oxidative stress in vivo and suppresses brain atrophy and decline in learning and memory associated with aging [1,2,3,4]. In this study, we investigated the effects of coffee polyphenols (CPP or chlorogenic acid) and milk fat globule membrane (MFGM) in the prevention of brain aging in senescence-accelerated mouse prone 8, SAMP8, which shows cognitive decline and the accumulation of amyloid-β in the brain at a relatively early stage. The SAM strain was developed by the selective inbreeding of AKR/J mice at Kyoto University, an inbred strain group consisting of senescence-prone relatives (SAMP) and senescence-resistant relatives (SAMR) [5]. 

Coffee contains nine CPPs, which mainly consist of caffeoyl quinic acid (CQA), feruloyl quinic acid (FQA), and dicaffeoyl quinic acid (diCQA); chlorogenic acid is the general term for these ester compounds of cinnamic acid derivatives and quinic acids [6]. CPP has been studied because it exhibits important pharmacological effects [7,8,9]. In this study, we aimed to investigate the neuroprotective effects of CPP in the hippocampus and cerebral cortex during aging, including its effects and mechanisms on cognitive function.

MFGM is a structure that takes the form of lipids, membrane-bound proteins, and glycans that surround the adipocyte secreted during mammalian lactation. Buttermilk, which is produced during butter production, is rich in MFGMs, and many studies have been conducted on the food functionality of MFGMs, including their importance in cognitive development and improvement of motor function [10,11,12]. 

Although coffee is often drunk with milk, no research has been conducted so far to explore its collaborative functionality. However, when considering each functionality of CPP and MFGM, we thought it would be worthwhile to examine the functionality of coffee with milk on cognitive function and aging. Therefore, we examined the effects of CPP and MFGM alone and when combined using SAMP8.

## 2. Results

### 2.1. Reduction in Mortality Due to CCP or MFGM Intake

SAMP8 mice that were two months of age were fed diets containing 2% CPP and 1% MFGM for one month or eight months, and brain tissue from young (3-month-old) and old (10-month-old) mice was collected (Figure 1). In the long-term experiment, some of the mice in the control group of SAMP8 died between 5 and 7 months, but a decrease in the mortality rate was observed in mice ingesting CCP and/or MFGM (Figure 2). A comparison of the survival rates through a log-rank test showed that the survival rate was significantly increased in the CD + MC group. The average intake was 2 mg/g for CPP and 1 mg/g for MFGM, which was based on the food intake and body weight of 9-month-old mice (Table 1).

### 2.2. Evaluation of Brain Function by Novel Object Recognition Test

In the memory retention trials at 9 months of age, the ratio of search time for novel objects to total search time was calculated as an index of cognitive function. As a result of multiple comparison tests among the SAMP8 groups, cognitive function was significantly improved in the CD + M, CD + C, and CD + MC groups compared to the CD group (Figure 3). The results were equal to or better than that of SAMR1, which indicates normal aging.

### 2.3. Changes in the Expression of Inflammatory Genes in the Hippocampus and Cerebral Cortex

Changes in gene expression related to inflammation were examined in the hippocampus and cerebral cortex through real-time PCR. In the hippocampus (Figure 4), the expression of proinflammatory cytokines such as tumor necrosis factor (TNFα) and interleukin 1β (IL-1β) was significantly increased in the SAMP8-aged controls. However, the expression of TNFα was significantly suppressed in mice treated with CPP and/or MFGM. In addition, the expression of IL-1β was suppressed in CPP-fed SAMP8 old mice similarly to SAMR1 mice showing normal aging. Meanwhile, the expression of IL-4, an anti-inflammatory cytokine, was not significantly changed by CPP or MFGM. On the other hand, the expression of transforming growth factor β1 (TGF-β1), another anti-inflammatory cytokine, was increased in young SAMP8 mice that ingested CPP and/or MFGM, to the same level as SAMR1, and was significantly increased in aged SAMP8 mice that were fed CPP or both CPP and MFGM. In addition, lipocalin 2 (Lcn2) was significantly upregulated in the MFGM group at a young age. Lcn2 is mainly secreted from astrocytes and is known to have a role in regulating the development of brain damage [13].

In the cerebral cortex (Figure 5), TNF-α expression was significantly increased in the aged control SAMP8 group but suppressed to levels comparable to SAMR1 in the SAMP8 that were fed CPP and/or MFGM. The expression levels of IL-1β and IL-4 were not significantly changed by the intake of CPP or MFGM, but the expression of TGF-β1 was increased in the older group with MFGM intake. Lcn2 tended to be upregulated in the MFGM group at a young age. In addition, the effect of CPP or MFGM ingestion on lysosomal enzyme cathepsin was examined in the cerebral cortex. The expression of cathepsin B (CtsB) tended to increase with age in the control group, but it was significantly decreased in the old mice fed CPP or both CPP and MFGM in the cerebral cortex. No such decrease was observed for cathepsin S (data not shown).

### 2.4. Changes in Expression of Genes Related to Cognitive Function and Lifespan

TGF-β1, which was found to be upregulated in the hippocampus and cortex (Figure 4 and Figure 5), has been reported as promoting the phosphorylation of the cAMP-responsive element binding (CREB) protein [14] and playing an important role in neurogenesis in the hippocampus after maturation [15]. Therefore, to examine the effects of CPP and MFGM intake on brain function, we focused on CREB, brain-derived neurotrophic factor (BDNF), and repressor element 1-silencing transcription factor (REST). REST has been highlighted as a gene that is associated with lifespan [16].

The results show that CREB expression was significantly increased in the hippocampus of the young CPP group, and its level was higher than that of the control SAMR1 (Figure 6). No significant change was observed in BDNF. The expression of REST increased in the aged hippocampus of the SAMP8 control group, while it was elevated from the young hippocampus of the CPP and MFGM groups. Its level was higher than that of SAMR1, the reference group. In the cortex, CREB expression increased with age, as did SAMR1. REST was shown to be different in the cortex than in the hippocampus, where CPP and MFGM intake did not increase the expression of young age SAMP8. Although REST expression increased with aging in the control and MFGM groups, the increase was suppressed by CPP intake.

### 2.5. Effects of CPP and MFGM on Microglial Cells

We examined the changes in cultured microglial cells (BV2) when they were treated with lipopolysaccharide (LPS) or IL-4 in the presence of CPP or MFGM. Here, 5-CQA was used as a representative CPP. When LPS was applied, the expressions of inducible nitric oxide synthase (iNOS), IL-1β, TNFα, and Lcn2 were drastically increased. CPP did not cause significant changes at any concentration for iNOS, IL-1β, TNFα, and Lcn2 (Appendix A). While it was found that CPP increased the expression of REST in BV cells, CREB expression was unchanged (Appendix A). On the other hand, when BV2 cells were activated with IL-4, TGF-β1 and IL-4 expression was enhanced by CPP (Appendix A). MFGM increased TGF-β1 expression but had no significant effect on LPS-activated BV2 cells and no enhancing effect on IL-4-activated BV2 cells (data not shown).

### 2.6. Effects of Systemic Circulation on Inflammation in the Brain

To examine whether the peripheral inflammatory response affects the increase in TNFα in the brain, we measured endotoxins in the serum of old mice. There were no positive data in each group (data not shown). The results showed that endotoxins in the serum of old mice were not elevated. 

## 3. Discussion

We examined the anti-aging effects of CPP and MFGM in SAMP8 mice and found that they reduced mortality and preserved long-term memory. In order to clarify the mechanism of the anti-aging effect, we examined changes in the expression of genes thought to be related to aging and brain function in the hippocampus and cerebral cortex. The results showed that the expression of REST was different between the hippocampus and the cerebral cortex, suggesting that its role differs in the brain.

SAMP8 is known to be a model of age-dependent neuroinflammation [17,18]. In fact, in the aged cerebral cortex of SAMP8, the expression of TNFα, a proinflammatory cytokine, was higher than in aged SAMR1, suggesting that inflammation in the brain may contribute to accelerated brain aging in SAMP8. In addition, the expression of CtsB was also higher in aged SAMP8 than in aged SAMR1 (Figure 5). CatB is involved in the induction of proinflammatory responses, resulting in cognition impairment [19,20]. However, the expression of TNF-α was decreased in the cerebral cortex of CPP-treated mice (Figure 5). The decrease was thought to suppress inflammation in the brain. REST, which was elevated in the cortex of aged mice, was suppressed by CPP ingestion (Figure 6). REST regulates many genes and signaling pathways involved in inflammatory processes [21], suggesting that the suppression of inflammation does not require an increased expression of REST in the cortex during aging (Figure 7A). That is, CPP and MFGM may suppress inflammation in the aging brain.

It is thought that only a small amount of CPP crosses the blood–brain barrier and is taken into the brain (private communication). Although millimolar-order CPP has been reported as suppressing the expression of TNF-α in microglia induced by LPS [22], micromolar-order CPP enhanced TGF-β1 in microglia induced by IL-4 (Appendix A). Since it is unlikely that CPP will actually reach millimolar concentrations in the brain, the direct anti-inflammatory effect of CPP may not be strong. In addition to the inflammation of the brain itself, chronic systemic inflammation is thought to be involved in the increase in inflammation in the brain with aging [23,24]. CPP and MFGM might suppress the increase in neuronal inflammation by regulating systemic inflammation, but no endotoxin was detected in the serum, at least in the aged mice of each group. Since a relationship between serum cytokines and longevity has been reported [25,26], it may be necessary to investigate the effects of peripheral cytokines in mice fed CPP and MFGM in the future. SAMP8 is also known to accumulate amyloid-β in the brain from a relatively early age [27], but in the present study there were no significant differences in the amount of insoluble amyloid-β in the cerebral cortex of each group (data not shown).

On the other hand, in the hippocampus of SAMP8, the expression of REST was enhanced by CPP and MFGM ingestion at a young age. Since the increased expression of REST is associated with the induction of neuronal differentiation [28], it is possible that CPP and MFGM ingestion may contribute to the suppression of memory decline. In addition, CREB expression was increased in the young SAMP8 hippocampus (Figure 6) and TGF-β1 enhances CREB phosphorylation [14], suggesting that CPP and MFGM play a role in enhancing brain function by phosphorylating and activating CREB, whose expression is increased in the hippocampus (Figure 7B). Since it has been suggested that the increased expression of CREB may be a factor in the increased expression of REST [29], it is possible that the increased expression of CREB results in the increased expression of REST in the hippocampus. Taken together, the increased expression of CREB, TGF-β1, and REST in the young hippocampus by CPP and MFGM may be important for the maintenance of cognitive function.

## 4. Materials and Methods

### 4.1. Animals and Reagents

SAMP8 mice were used as the experimental animals. Four- to five-week-old male mice were purchased from Japan SLC, and three mice per cage were kept under a 12 h light/dark cycle (8:00–20:00 light period) at room temperature of 23 ± 1 °C and humidity of 55 ± 5%. Acclimation rearing was conducted until the animals reached 2 months of age. Feed (AIN-93G, Oriental Yeast Co., Ltd. Tokyo, Japan) and drinking water were ad libitum, and CPP (C) and MFGM (M) were administered to the mice as a solid diet, with AIN-93G (CD) as the base feed. 

CPP were extracted from green coffee beans with hot water and spray dried. The composition of CPP was analyzed by high-performance liquid chromatography. The total polyphenol content was 38.0% and composed of various polyphenols as follows: 8.1% 3- CQA, 8.1% 4-CQA, 11.0% 5-CQA, 1.4% 3-FQA, 1.4% 4-FQA, 1.9% 5-FQA, 2.5% 3,4-diCQA, 1.4% 3,5-diCQA, and 2.4% 4,5-diCQA. The CPP preparation contained no caffeine. MFGM was purchased from LECICO GmbH (Lipamin M20: Uelzen, Germany), and was prepared from a butter serum that contained concentrated milk phospholipids [30]. 

The animal care and experiments were conducted in accordance with the University of Shizuoka’s guidelines for animal experiments. For SAMP8, the number of mice per group was 21, and 4 groups were set up. Mice were divided into a control group (CD), a group receiving food containing 2% CPP (CD + C), a group receiving food containing 1% MFGM (CD + M), and a group receiving food containing both CPP and MFGM (CD + MC). SAMR1 mice showing normal aging were given CD as a reference group (*n* = 12).

After a month of acclimation, 9 mice per group were dissected after 1 month of feeding (3 months of age) for SAMP8, and 6 mice were dissected for SAMR1 to make samples of young mice. Then, the experiment continued and the surviving mice were dissected after 9 months of feeding (10 months of age) and used as samples of old mice. Food intake, body weight gain, and survival time were measured during this period. Before dissection, we conducted a novel object recognition test to examine the effects on brain function. All experimental protocols were approved by the University of Shizuoka’s Laboratory Animal Care Advisory Committee (approval No. 195242, 13 June 2019) and were in accordance with the guidelines of the US National Institutes of Health for the care and use of laboratory animals.

### 4.2. Novel Object Recognition Test

In order to familiarize the mice with the gray acrylic apparatus (50 cm long × 50 cm wide × 40 cm high), each mouse was allowed to explore the apparatus for 5 min. The next day, two identical objects (triangular pyramids) were placed in the apparatus as a training test, and the contact time with each object was recorded for 5 min. A day later, one of the objects was replaced with a new object (quadrangular prism), and the contact time with each object was recorded for 5 min. In the memory retention trial, the ratio of search time for the new object to the total search time was calculated and used as an index of cognitive function.

### 4.3. Quantitative Real-Time Reverse Transcription PCR (qRT-PCR)

The mice were anesthetized with isoflurane and blood was collected from the abdominal aorta. The brain was carefully dissected, and the hippocampus and cerebral cortex were immediately frozen. Real-time PCR was performed on the brain samples to compare the expression changes of each gene. Total RNA was extracted from tissues using a purification kit (NucleoSpin^®^ RNA, 740955, Takara Bio Inc., Kusatsu, Japan) in accordance with the manufacturer’s protocol. The obtained RNA was converted to cDNA using the PrimeScript™ RT Master Mix kit (RR036A, Takara Bio Inc.). A real-time quantitative PCR analysis was performed using the PowerUp™ SYBR™ Green Master Mix (A25742, Applied Biosystems Japan Ltd., Tokyo, Japan) and automated sequence detection systems (StepOne, Applied Biosystems Japan Ltd.). Relative gene expression was measured by previously validated primers for TNFα [31], IL-1β [32], IL-4 [33], TGF-β1 [34], Lcn2 [35], CtsB [36], CREB [37], BDNF [38], REST [39], and iNOS [40] genes. The primer sequences are set out in Table 2. CDNA derived from transcripts encoding β-actin was used as the internal control.

### 4.4. Effects of CPP and MFGM on Microglia

BV2 cells (Elabscience Biotechnology Inc., Houston, TX, USA) were counted and seeded at 4 × 10^5^ cells/500 μL/well in 24-well plates and incubated for 24 h at 37 °C under 5% CO_2_. Culture medium was MEM with Earle’s Salts and L-Gln (21442-25, Nacalai Tesque, Inc., Kyoto, Japan), with 10% fetal bovine serum (FBS, Mediatech Inc., Tokyo, Japan) and 1% penicillin-streptomycin (Nacalai Tesque Inc.). Then, 5-CQA (C3878, Sigma-Aldrich Co., LLC, Tokyo, Japan) was used as CPP; 5-CQA (0200 µM in MEM) or MFGM (dissolved in MEM containing 1% DMSO to be 00.01%) was added and incubated for 1 h. LPS (O55:B5, L2880, 100 ng/mL, Sigma-Aldrich) or IL-4 (20 ng/mL, FUJIFILM Wako Chemical Co., Kanagawa, Japan) was added and incubated for 24 h. The cells were treated with 0.5 *w*/*v*% trypsin-EDTA solution (FUJIFILM Wako Chemical Co.). The collected cells were used for qRT-PCR measurements.

### 4.5. Measurement of Endotoxin in the Serum of Aged Mice

Blood collected from the abdominal aorta was centrifuged using Capiject (CJ-AS, Terumo Co., Tokyo, Japan). The obtained serum was stored at −80 °C until the time of use. Endotoxin levels were measured using an ELISA kit (Endospecy, Seikagaku Co., Tokyo, Japan), which is based on the activation and gelation of limulus amebocyte lysate by endotoxin derived from Gram-negative bacteria.

### 4.6. Statistical Analyses

The results are expressed as the mean ± SEM. A statistical analysis was performed using one-way ANOVA, and statistical significance was set at *p* < 0.05. Confidence intervals and significance of differences in means were estimated by using the Tukey–Kramer significant difference method or Fisher’s least significant difference test. After calculating survival rates using the Kaplan–Meier method, the difference in survival rate was tested using the log-rank test. 

## 5. Conclusions

The ingestion of CPP and MFGM extended the lifespan of SAMP8 mice and retained their cognitive function. This was partly due to the suppression of inflammation in the brain of old age mice by CPP and MFGM intake. The endotoxin levels were not elevated in the serum of aged mice. The increased expression of CREB, TGF-β1, and REST in the hippocampus at a young age may be important for the maintenance of cognitive function. It is suggested that the addition of milk to coffee may enhance CPP’s functionality.

## Figures and Tables

**Figure 1 ijms-23-05832-f001:**
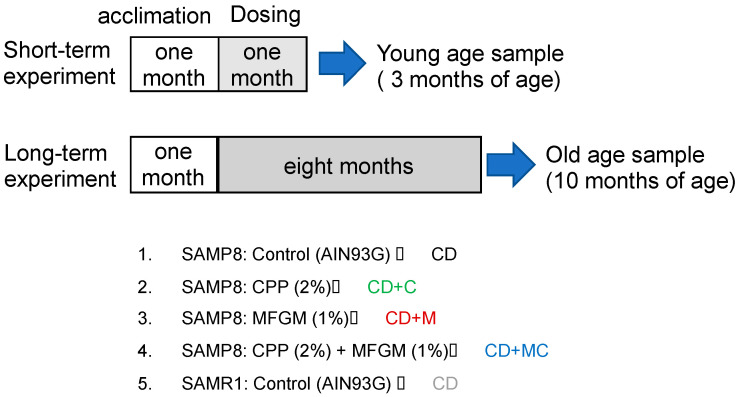
Experimental protocol.

**Figure 2 ijms-23-05832-f002:**
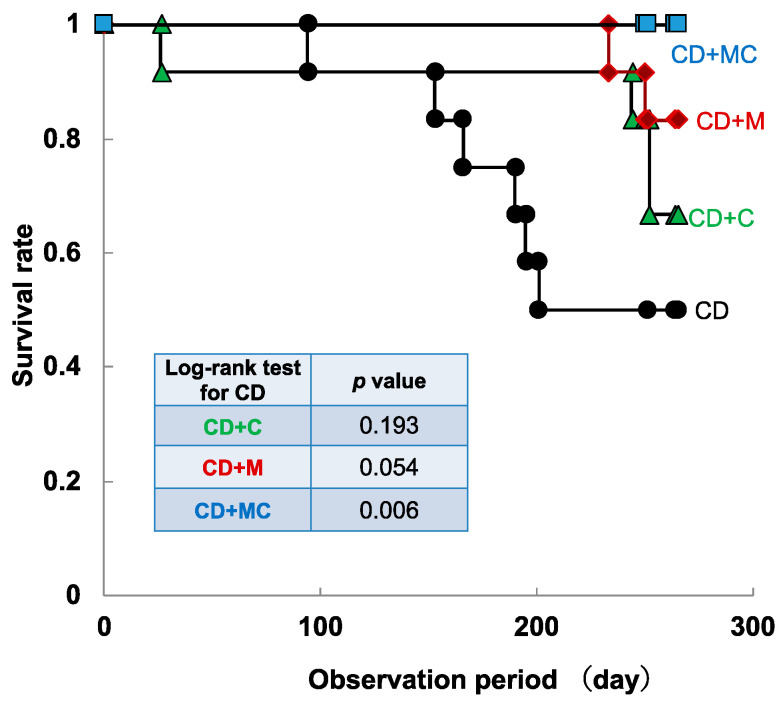
Reduction in mortality of SAMP8 due to CCP and MFGM intake. In the long-term experiments, SAMP8 mice (*n* = 12/group) were fed diets containing CPP and/or MFGM ad libitum.

**Figure 3 ijms-23-05832-f003:**
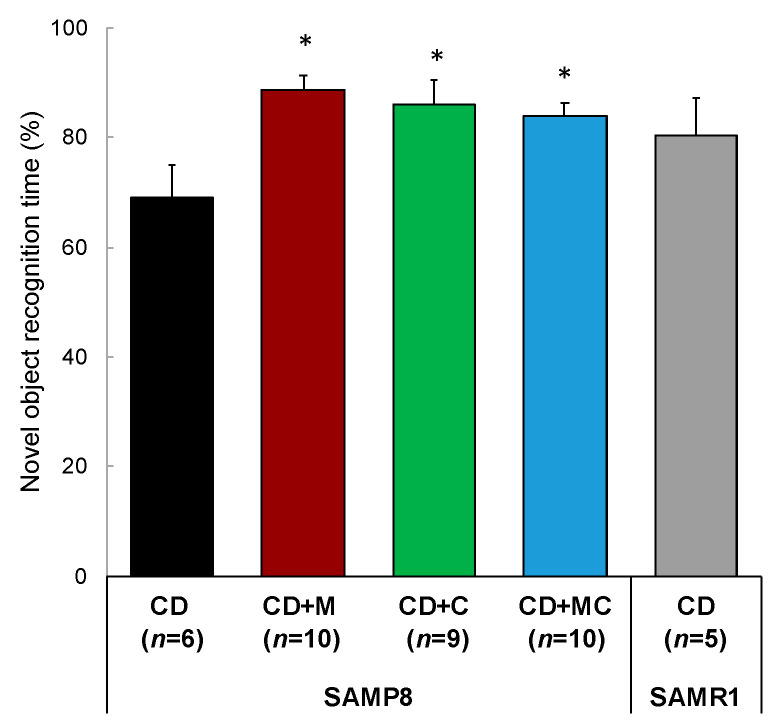
Improved memory retention due to CCP and MFGM intake. Each value represents the mean ± SEM. Asterisks denote significant differences relative to SAMP8 control (*, *p* < 0.05 Tukey–Kramer significant difference method).

**Figure 4 ijms-23-05832-f004:**
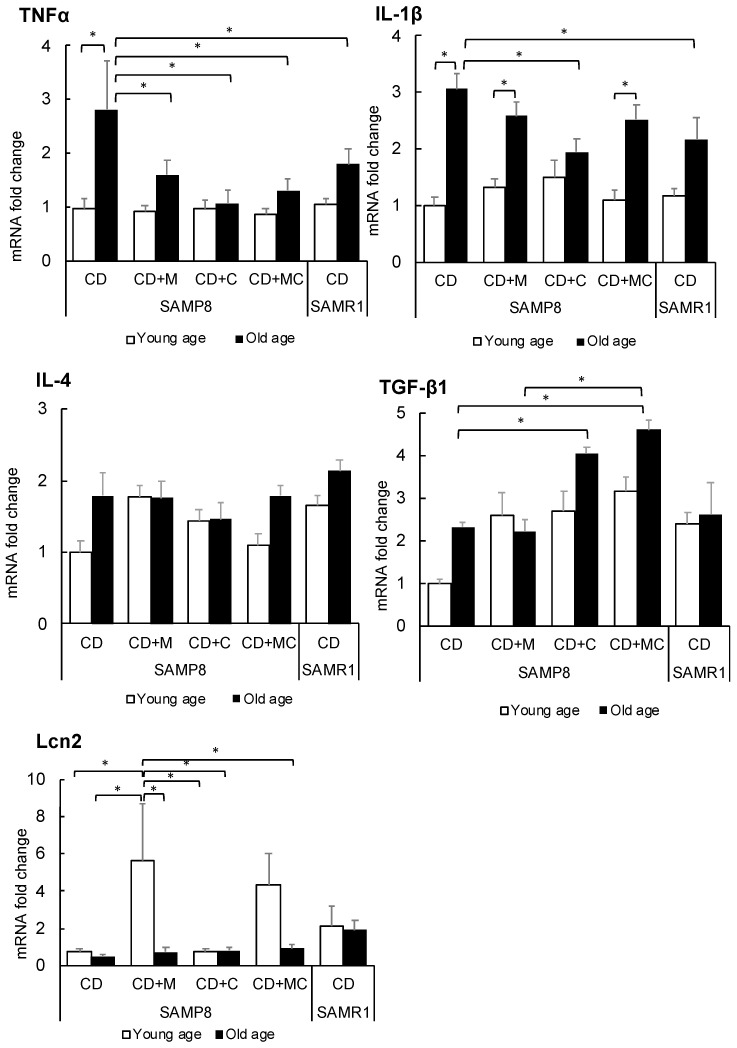
Changes in the expression of inflammatory genes in the hippocampus. Each value represents the mean ± SEM (*n* = 6). Asterisks denote significant differences (*, *p* < 0.05 Fisher’s least significant difference test).

**Figure 5 ijms-23-05832-f005:**
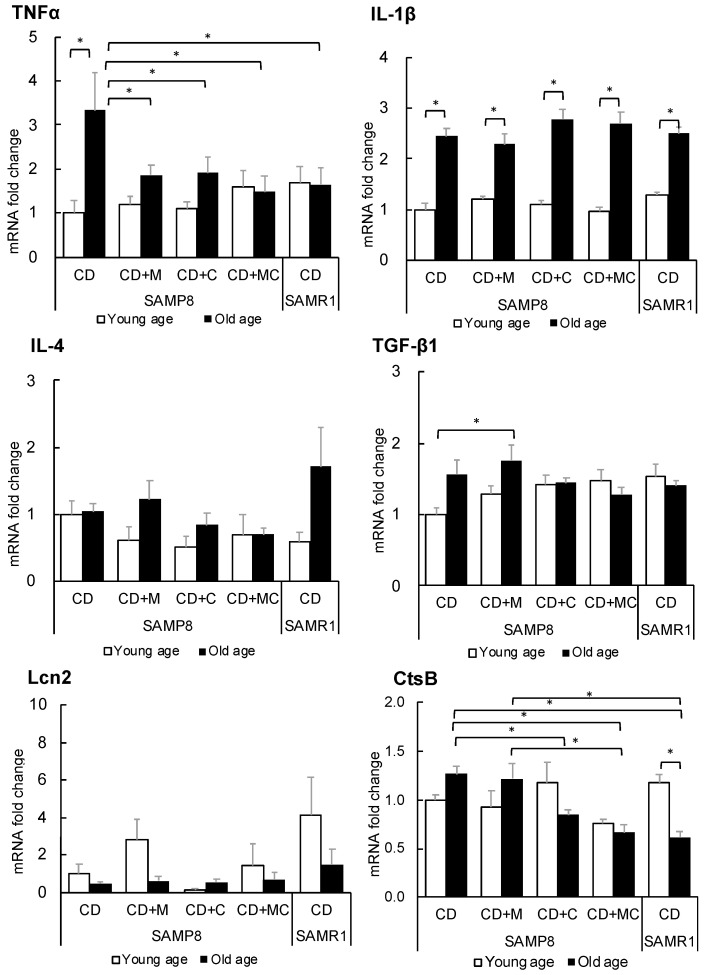
Changes in the expression of inflammatory genes in the cerebral cortex. Each value represents the mean ± SEM (*n* = 6). Asterisks denote significant differences (*, *p* < 0.05, Fisher’s least significant difference test).

**Figure 6 ijms-23-05832-f006:**
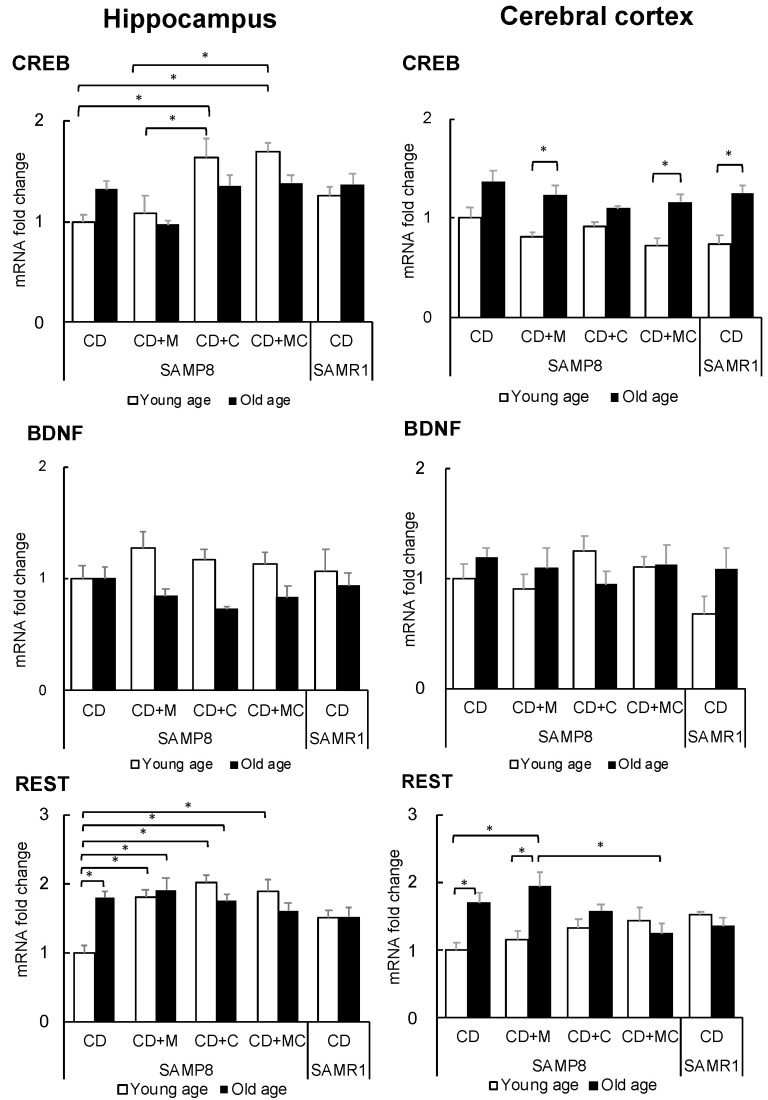
Expression of CREB, BDNF, and REST in the hippocampus and cerebral cortex. Each value represents the mean ± SEM (*n* = 6). Asterisks denote significant differences (*, *p* < 0.05, Fisher’s least significant difference test).

**Figure 7 ijms-23-05832-f007:**
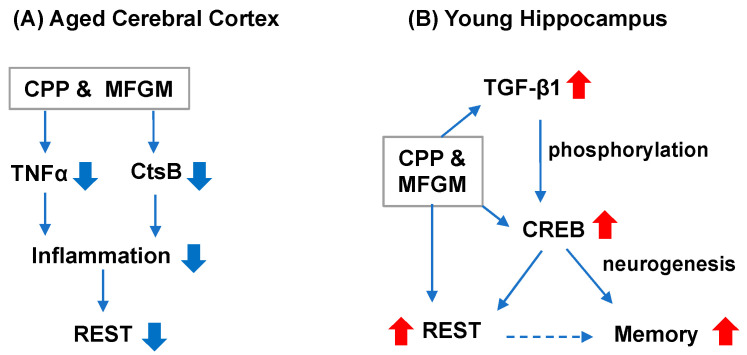
Effects of CPP and MFGM in the cerebral cortex of aged SAMP8 mice (**A**), and in the hippocampus of young mice (**B**). Blue arrows indicate a decrease and red arrows indicate an increase.

**Table 1 ijms-23-05832-t001:** Ingestion of CCP or MFGM.

Group	Mice Number	Body Weight (g)	Feed Intake (g/Mouse/Day)	Average Intake of CPP and/or MFGM (mg/g Body Weight/Day)
CD	6	38.9 ± 5.3	3.65 ± 0.27	
CD + M	10	37.0 ± 4.9	3.53 ± 0.16	1 mg MFGM
CD + C	9	43.3 ± 6.1	3.98 ± 0.23	2 mg CPP
CD + MC	10	39.5 ± 5.1	4.39 ± 0.28	1 mg MFGM and 2 mg CCP

Based on 9-month-old data. Each value represents the mean ± SEM.

**Table 2 ijms-23-05832-t002:** Sequence of primers used in the qRT-PCR.

Gene	Forward Sequence	Reverse Sequence	Ref.
TNF-α	CTGTCTACTGAACTTCGGGGTGAT	GGTCTGGGCCATAGAACTGATG	[31]
IL-1β	GCAACTGTTCCTGAACTCAACT	ATCTTTTGGGGTCCGTCAACT	[32]
IL-4	CAGCTAGTTGTCATCCTGCTCTTC	GCCGATGATCTCTCTCAAGTGA	[33]
TGF-β1	CAAGGGCTACCATGCCAACT	GTACTGTGTGTCCAGGCTCCAA	[34]
Lcn2	TACAATGTCACCTCCATCCTGG	TGCACATTGTAGCTCTGTACCT	[35]
CtsB	CTGCTGAAGACCTGCTTA	AATTGTAGACTCCACCTGAA	[36]
CREB	GAGAGCTGGTATGTCAGGAATG	CCAGAAGAGATGCAGGAGAAAG	[37]
BDNF	TACTTCGGTTGCATGAAGGCG	GTCAGACCTCTCGAACCTGCC	[38]
REST	ATCGGACGCGGGTAGCGAG	GGCTGCCAGTTCAGCTTTCG	[39]
iNOS	GGCAAACCCAAGGTCTACGTT	GAGCACGCTGAGTACCTCATTG	[40]
β-actin	TGACAGGATGCAGAAGGAGA	GCTGGAAGGTGGACAGTGAG

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
