# Peer review of "Coffee Polyphenol, Chlorogenic Acid, Suppresses Brain Aging and Its Effects Are Enhanced by Milk Fat Globule Membrane Components"

_ijms, 2022, doi:10.3390/ijms23105832_

Round 1

Reviewer 1 Report

The authors present a provocative study, showing:

(1) a survival benefit for CPP+MFGM group, trend towards increased survival for CPP mad MFGM groups alone;

(2) less cognitive decline (as measured by search time for novel objects) in all treated groups (positively confounded by survival bias in the control group; most impaired mice might already have died by 9 months);

(3) pro-inflammatory cytokines less increased (mRNA expression) in aged treated groups compared to young.

The animal experiments are clearly described and the findings and underlying question will be of wide interest.  

Major suggestion for improvement: Move section 2.5 (including Figs 7,8 & 9 (as well as 10c panel) to supplemental (or remove), the section and figures add little to the manuscript. While exploring the effects of CQA on cultured microglia is merited, the largely negative, and non-dose response, nature of the findings detract from the main findings. A brief summary of the results could be combined with the negative endotoxin findings of section 2.6.  

Minor suggestions: What are the error ranges in table 1 (SEM?). Please also provide n-numbers in the legend.  

Language points of note: I feel the first sentence of abstract should read: "Mice feed with..." rather than "Mice feeding on..."

Author Response

Response to Reviewer 1

The authors present a provocative study, showing:

(1) a survival benefit for CPP+MFGM group, trend towards increased survival for CPP and MFGM groups alone;

(2) less cognitive decline (as measured by search time for novel objects) in all treated groups (positively confounded by survival bias in the control group; most impaired mice might already have died by 9 months);

(3) pro-inflammatory cytokines less increased (mRNA expression) in aged treated groups compared to young.

The animal experiments are clearly described and the findings and underlying question will be of wide interest.  

Thank you very much for reviewing our manuscript.

Major suggestion for improvement: Move section 2.5 (including Figs 7,8 & 9 (as well as 10c panel) to supplemental (or remove), the section and figures add little to the manuscript. While exploring the effects of CQA on cultured microglia is merited, the largely negative, and non-dose response, nature of the findings detract from the main findings.

Figures 7, 8 and 9 has been moved to the Supplement. Figure 10c was removed.

A brief summary of the results could be combined with the negative endotoxin findings of section 2.6.  

The results of the endotoxin have been added to the abstract and conclusion (line 24, and line 31).

Minor suggestions: What are the error ranges in table 1 (SEM?). Please also provide n-numbers in the legend.  

A description of the number of samples and errors was added to Table 1.

Language points of note: I feel the first sentence of abstract should read: "Mice feed with..." rather than "Mice feeding on..."

Thank you for pointing this out. This sentence has been corrected (line 14).

Reviewer 2 Report

The overall goal of this work was to examine the effects of coffee polyphenols (CPP, chlorogenic acid) and milk fat globule membrane (MFGM) on brain aging. The authors used a mouse model which has an accelerated aging phenotype. It was observed that the mice had longer survival periods with CPP and MFGM ingestion but not CPP or MFGM alone. Memory, as tested by novel object recognition was also improved; although this is not an all encompassing cognitive test with regards to spatial memory. Some changes to inflammatory cytokine expression were observed and changes to the expression of REST and CREB. The authors also examined microglial phenotypes in BV2 cells activated with various cytokines and treated with CPP/MFGM. The cell culture results are not convincing overall and slightly confusing. Furthermore, the field no longer accepts the terms M1/M2 phenotypes of microglia in general. The accepted terminology is disease associated microglia. As such this indicated microglial phenotypes that are associated with disease. The article is also missing background regarding the animal model and what SAMR1 is. I think the graphs would overall be more informing if they were organized differently. For example grouping young animals together and old animals together. Its also unclear how ANOVA where ran and what samples were compared in the ANOVA. There are no n reported for the old mice. This information should be in the figure legend for each figure. Furthermore, testing endotoxin levels systemically does not empirically rule out systemic inflammation. You can either test cytokine levels in serum or state endotoxin levels were not elevated; but you can not say no systemic inflammation was observed.  

Author Response

Response to Reviewer 2

The overall goal of this work was to examine the effects of coffee polyphenols (CPP, chlorogenic acid) and milk fat globule membrane (MFGM) on brain aging. The authors used a mouse model which has an accelerated aging phenotype. It was observed that the mice had longer survival periods with CPP and MFGM ingestion but not CPP or MFGM alone. Memory, as tested by novel object recognition was also improved; although this is not an all encompassing cognitive test with regards to spatial memory. Some changes to inflammatory cytokine expression were observed and changes to the expression of REST and CREB.

The authors also examined microglial phenotypes in BV2 cells activated with various cytokines and treated with CPP/MFGM. The cell culture results are not convincing overall and slightly confusing.

Thank you very much for reviewing our manuscript.

Furthermore, the field no longer accepts the terms M1/M2 phenotypes of microglia in general. The accepted terminology is disease associated microglia. As such this indicated microglial phenotypes that are associated with disease.

Thank you so much for your valuable suggestion. We have revised section 2.5 and discussion.

The article is also missing background regarding the animal model and what SAMR1 is.

Background on the SAM mouse has been added to the introduction (line 41-43).

I think the graphs would overall be more informing if they were organized differently. For example grouping young animals together and old animals together.

The purpose of this study is to compare the extent to which changes occurred at old age compared to young age in each group. If the groups were divided into young and old, it would be rather difficult to compare the the effects of CPP or MFGM intake between young and old group.

Its also unclear how ANOVA where ran and what samples were compared in the ANOVA. There are no n reported for the old mice. This information should be in the figure legend for each figure.

In this study, the commonly used multiple comparison test was used. That is, an analysis of variance (ANOVA) was first performed on the young and old groups that received CPP or MFGM, and then a significance difference test among groups was conducted using Fisher's least significant difference test or Tukey-Kramer significant difference method.

The number n was added in Table 1.

In all Figures, n numbers are written. In Figures 4 through 6, n number for both young and old mice are 6.

Furthermore, testing endotoxin levels systemically does not empirically rule out systemic inflammation. You can either test cytokine levels in serum or state endotoxin levels were not elevated; but you can not say no systemic inflammation was observed.  

We revised it (line 163).

Round 2

Reviewer 2 Report

All comments have been addressed